



Atmospheric
Measurement
Techniques

# A Bayesian parametric approach to the retrieval of the atmospheric number size distribution from lidar data

**Alberto Sorrentino**[1], **Alessia Sannino**[2], **Nicola Spinelli**[2], **Michele Piana**[1], **Antonella Boselli**[3], **Valentino Tontodonato**[5], **Pasquale Castellano**[5], **and Xuan Wang**[4]

[1]Dipartimento di Matematica, Università di Genova, Genova, Italy
[2]Dipartimento di Fisica, Università di Napoli Federico II, Napoli, Italy
[3]CNR-IMAA, Potenza, Italy
[4]CNR-SPIN, Napoli, Italy
[5]ALA Srl Advanced Lidar Applications, Napoli, Italy

**Correspondence:** Alberto Sorrentino (sorrentino@dima.unige.it)

**Abstract.** We consider the problem of reconstructing the number size distribution (or particle size distribution) in the atmosphere from lidar measurements of the extinction and backscattering coefficients. We assume that the number size distribution can be modeled as a superposition of log-normal distributions, each one defined by three parameters: mode, width and height. We use a Bayesian model and a Monte Carlo algorithm to estimate these parameters. We test the developed method on synthetic data generated by distributions containing one or two modes and perturbed by Gaussian noise as well as on three datasets obtained from AERONET. We show that the proposed algorithm provides good results when the right number of modes is selected. In general, an overestimate of the number of modes provides better results than an underestimate. In all cases, the $PM_1$, $PM_{2.5}$ and $PM_{10}$ concentrations are reconstructed with tolerable deviations.

## 1 Introduction

Lidar (light detection and ranging) is a remote sensing technique similar to radar (radio detecting and ranging) which uses light in the form of short laser pulses to invest a target and obtain, through elastic and inelastic scattering processes, information on the target properties as a function of the distance from the lidar system.

In the atmospheric application, lidar systems can be used to obtain spatially resolved information about the optical properties of the atmospheric aerosols (desert dust, volcanic ash, smog and many other types of substances) (Giannakaki et al., 2010; Ritter et al., 2018; Lee and Wong, 2018; Stelitano et al., 2019; Chazette, 2020) over a distance of several kilometers and with high spatial and temporal resolutions. Advanced, multiwavelength lidar systems are capable of giving information on the spectral dependence of the aerosol optical properties and can therefore be used to obtain information on the aerosol microphysical properties (Weitkamp, 2006; Pérez-Ramírez et al., 2013; Granados-Muñoz et al., 2014; Giannakaki et al., 2016; Müller et al., 2016; Chemyakin et al., 2016; Ortiz-Amezcua et al., 2017; Molero et al., 2020).

However, information on the microphysical properties of the atmospheric aerosols is seldom obtained using the lidar signal alone. This information, which is essential for a complete aerosol characterization useful to understand their effect on climate, is instead frequently obtained through the synergistic use of in situ instruments; incidentally these measurements also allow a validation of the lidar retrievals, but only for those values that are closest to the ground and for a particular aerosol typology (Saharan dust, biomass burning aerosol, etc.); alternatively, validation can be done using synthetic data (Alados-Arboledas et al., 2011; Di Girolamo et al., 2012; Veselovskii et al., 2013; Osterloh et al., 2013; Samaras et al., 2015; Whiteman et al., 2018).

In order to retrieve the microphysical properties of the aerosol from lidar measurements, two inverse problems must

be solved in sequence: in the first inverse problem, one uses the measured backscattered power to obtain an estimate of the aerosol optical parameters; in the second problem, one uses the estimated optical parameters (at different wavelengths), derived from lidar observations, to obtain an estimate of the number size distribution, i.e., the density of particles as a function of the particle size. This latter problem is particularly challenging because of the limited number of data due to the practical problem of measuring at many different wavelengths. Moreover, from a mathematical point of view, the microphysical parameters are generally derived from the optical ones through integral equations that cannot be solved analytically and whose numerical solution leads to a so-called ill-posed problem. This last is characterized by a strong sensitivity of the solution from the input data uncertainties and by the non-uniqueness of the solution. Remarkably, from a mathematical viewpoint there can be several ways to overcome ill-posedness; however, not all of them actually reflect realistic physical conditions. In addition, numerical studies showed that a poor selection of the constraints can affect the quality of the solution and compromise the microphysical retrieval in spite of the strength of the regularization algorithm. Therefore, in order to obtain stable and physically acceptable solutions, mathematical and physical constraints variously combined with regularization methods are applied (Müller et al., 1999; Böckmann, 2001; Veselovskii et al., 2002; Böckmann et al., 2005; Kolgotin and Müller, 2008).

In the past decade, a number of studies have focused on the retrieval of the microphysical aerosol parameters from multi-wavelength lidar measurements using the standard "$3\beta + 2\alpha$" configuration (i.e., the measurement of the backscattering coefficient at three wavelengths and the extinction coefficient at two wavelengths) (Müller et al., 1998, 2001; Wandinger et al., 2002; Müller et al., 2003; Murayama et al., 2004; Müller et al., 2006; Tesche et al., 2008; Noh et al., 2009; Balis et al., 2010; Alados-Arboledas et al., 2011; Navas-Guzman et al., 2013; Nicolae et al., 2013; Sawamura et al., 2014; Chemyakin et al., 2014; Burton et al., 2016; Tesche et al., 2019; Pérez-Ramírez et al., 2019, 2020; McLean et al., 2021). Most of these studies refer mainly to data from ground-based elastic and Raman lidars working at 355, 532 and 1064 nm. However, the validation of such retrievals is a challenging task due to the unavailability of direct collocated measurements.

Other studies have compared the "$3\beta + 2\alpha$" lidar retrievals to the AERONET (Aerosol Robotic Network) retrievals (Holben et al., 1998; Veselovskii et al., 2009; Sawamura et al., 2014). Nevertheless, AERONET retrieval of columnar volume concentrations, assuming the aerosol is uniformly mixed throughout the boundary layer, is not directly comparable to vertically resolved lidar results. Intercomparison studies between "$3\beta + 2\alpha$" lidar retrievals and AERONET retrievals or ground-based in situ measurements are not suitable to properly evaluate the performance of the lidar microphysical retrievals obtained for different altitudes.

Despite these difficulties, the possibility of characterizing the atmospheric particulate using only the lidar instrument would be very advantageous, and for these reasons it is currently a much studied topic (Di et al., 2018a; Kolgotin et al., 2016; De Rosa et al., 2020).

Following the state of the art, we retrieve the particle size distribution from "$3\beta + 2\alpha$" lidar system parameters. In order to mitigate the ill-posedness, we adopt a parametric model for the number size distribution. Several authors referred to this shape as triangular, such as in (Veselovskii et al., 2004), defined on equidistant or a logarithmic-equidistant grid for the effective radius of the particles. In agreement with Di et al. (2018b) and Sun et al. (2013) and with the standard AERONET inversion procedure (Dubovik et al., 2006), we find it more viable to work with log-normal distributions. In particular we reconstruct the particle size distribution as a superposition of a small number of log-normal distributions on a logarithmic interval of particle radius. In practice, this entails that the number of unknowns to be estimated is either three, in the unimodal case, or six, in the bimodal case; the problem is therefore overdetermined in the unimodal case and underdetermined in the bimodal case. In both cases, it is very useful to have the possibility of exploiting any information one might have a priori, for instance on the plausible interval range for the values of these unknowns; to this aim, we set up a Bayesian model where such a priori information can be coded in proper *prior* distributions, and we then use uniform priors in selected intervals. Then, because of the non-linearity of the problem, we adopt a Markov Chain Monte Carlo algorithm (Metropolis–Hastings) to approximate the posterior distribution of the parameters of interest. Monte Carlo methods have long been considered to be of little practical use due to their high computational cost; however, the steady growth of available computational power that characterized the last decades has made them largely usable in the applications. We reckon the proposed method features three main advantages with respect to the state of the art: (i) because it is based on a Bayesian model, it naturally provides uncertainty quantification on the estimated parameters, which is not always the case for competitors; (ii) because it makes use of a Monte Carlo algorithm, it does not get stuck in local minima like deterministic optimization algorithms often do; and (iii) for the same reason it can be easily generalized to include, for instance, a non-Gaussian distribution of the noise term.

This article is organized as follows: in the "Methods" section, we provide the mathematical formulation of the problem and a description of the Monte Carlo algorithm; in the following section, we analyze the results obtained for synthetic data using five exemplar cases; in the final section we briefly summarize our conclusions.

## 2   Methods

Lidar instruments measure the backscattered light power at wavelength $\lambda$, coming from distance $z$, given by the following equation:

$$P(\lambda, z) = \frac{C\beta(\lambda, z)}{z^2} \exp\left(-\int\limits_0^z \alpha(\lambda, x) \mathrm{d}x\right), \qquad (1)$$

where $\alpha(\lambda, z)$ and $\beta(\lambda, z)$ are the extinction and backscattering coefficients, respectively, and $C$ is a constant that depends on the instrument characteristics. Equation (1) can be looked at as an *inverse problem*, where one aims at recovering the extinction and backscattering coefficients from lidar data. By solving the inverse problem, (see Ansmann et al., 1990, 1992; Shcherbakov, 2007; Pornsawad et al., 2008; Garbarino et al., 2016; Denevi et al., 2017), one obtains extinction and backscattering coefficients at every altitude $z$ for a usually small set of $\lambda$. The present article is concerned with the subsequent problem of estimating the number size distribution from these indirect measures of $\alpha(\lambda, z)$ and $\beta(\lambda, z)$. In the following, we omit the dependence on $z$ because the problem can be solved independently at different altitudes.

### 2.1   Definition of the problem

The extinction and backscattering coefficients carry information on the number size distribution through the Mie scattering theory. Specifically, let $n(r)$ be the number size distribution; then, under the approximation of spherical homogeneous particles, the extinction and backscattering coefficients are given by

$$\alpha(\lambda) = \int\limits_{r_a}^{r_b} k_\alpha(r, \lambda, m) n(r) \mathrm{d}r \qquad (2)$$

$$\beta(\lambda) = \int\limits_{r_a}^{r_b} k_\beta(r, \lambda, m) n(r) \mathrm{d}r, \qquad (3)$$

where $r_a$ and $r_b$ are the lower and upper bounds for the particles' size, $m$ is the complex refractive index (CRI) of the target atmosphere, and $k_{\alpha/\beta}(r, \lambda, m)$ are integral kernels defined as follows.

$$k_\alpha(r, \lambda, m) = \frac{2\pi}{k^2} \sum_{n=1}^{\infty} (2n+1) \mathcal{R}(a_n + b_n) \qquad (4)$$

$$k_\beta(r, \lambda, m) = \frac{\pi}{k^2} \left| \sum_{n=1}^{\infty} (2n+1)(-1)^n (a_n - b_n)^2 \right|, \qquad (5)$$

where $a_n$ and $b_n$ are defined as follows:

$$a_n(x, m) = \frac{m\psi_n(mx)\psi_n'(x) - \psi_n(x)\psi_n'(mx)}{m\psi_n(mx)\xi_n'(x) - \xi_n(x)\psi_n'(mx)} \qquad (6)$$

$$b_n(x, m) = \frac{\psi_n(mx)\psi_n'(x) - m\psi_n(x)\psi_n'(mx)}{\psi_n(mx)\xi_n'(x) - m\xi_n(x)\psi_n'(mx)}, \qquad (7)$$

$\psi_n$ and $\xi_n$ being the Riccati–Bessel functions.

The problem we want to solve consists of retrieving the number size distribution $n(r)$ from a set of measurements $\{\alpha(\lambda_i)\}_{i=1,\ldots,I}$ and $\{\beta(\lambda_j)\}_{j=1,\ldots,J}$. By defining a data array $y = (\alpha(\lambda_1), \ldots, \alpha(\lambda_I), \beta(\lambda_1), \ldots, \beta(\lambda_J))$ and discretizing the possible values of $r$, we get an inverse problem of the form TS1

$$y = \mathbf{K}n + \varepsilon, \qquad (8)$$

where $n$ is a vector such that $n_i = n(r_i)$, $\mathbf{K}$ is the discretization of the integral kernels, and $\varepsilon$ is the noise affecting the data. Typical experimental values are such that $\alpha$ is measured at two wavelengths ($\lambda = 355, 532\,\mathrm{nm}$), and $\beta$ is measured at three wavelengths ($\lambda = 355, 532, 1064\,\mathrm{nm}$).

### 2.2   Parametric model

Solving the linear inverse problem defined by Eq. (8) with a reasonable discretization of the $r$ variable (say, at least 200 points) entails recovering a large number of unknowns from a very small dataset. One viable option is to reduce the number of unknowns by using a variable support function, such as in Osterloh et al. (2011). Here we take a different approach and assume that the number size distribution has a pre-defined shape that depends on a small number of parameters (Osterloh et al., 2013). Specifically, we assume that the number size distribution is the superposition of a small number $N$ of log-normal distributions.

$$n(r) = \sum_{i}^{N} \frac{h_i}{r} \exp\left(-(\log(r) - \log(\mu_i))^2 / 2\log(\sigma_i^2)\right) \qquad (9)$$

Each log-normal distribution is completely defined by three parameters: its mean $\mu$, its standard deviation $\sigma$ and its height $h$. The total number of parameters to be estimated is therefore $3N$, which is substantially smaller than the number of parameters for solving the unconstrained, linear case. However, the problem is now non-linear and cannot be solved with standard Tikhonov regularization. Indeed, we can rewrite Eq. (8) as

$$y = \sum_{i=1}^{N} h_i \mathbf{K}_m(\mu_i, \sigma_i) + \varepsilon, \qquad (10)$$

where $\mathbf{K}_m(\cdot, \cdot)$ represents non-linear functions of the mode and width of each component and can be derived by inserting Eq. (9) into Eqs. (2)–(3). We observe that the dependence on the CRI $m$ is indicated here by the subscript because we consider $m$ to be fixed and known throughout this study.

In this work, we set up a Bayesian model and apply a Monte Carlo technique to find the best parameters of unimodal ($N = 1$) and bimodal ($N = 2$) distributions, given a priori information and the data.

## 2.3 Bayesian approach

In the Bayesian framework, probability distributions are used to code our degree of knowledge on the values of unknown or unobservable quantities: perfect knowledge is represented by a probability distribution which is non-zero only in the correct value, and partial knowledge is represented by a probability distribution which assigns high probability to likely values and low probability to unlikely values. The Bayesian framework is useful to combine a priori information, i.e., information available before the data are collected, with the information content of the data: a priori information is coded in the so-called *prior* distribution, while the information content of the data is conveyed by the *likelihood* function.

Let us define the vector $x = \{\mu_i, \sigma_i, h_i\}_{i=1,\dots,N}$ containing all the unknown parameters. We denote its prior density as $p(x)$; in this work, we assume that individual parameters are independent of each other so that their joint density is the product of single densities. We also assume that no additional prior information is available and use uniform prior distribution for each of them in a suitable interval. This leads us to define the prior as

$$p(x) = \prod_{i=1}^{N} \mathcal{U}(\mu_i)\mathcal{U}(\sigma_i)\mathcal{U}(h_i),$$

where $\mathcal{U}()$ denotes the uniform distribution.

As far as the likelihood is concerned, we assume that the data are affected by Gaussian noise, which leads us to define the likelihood as

$$p(y|x) = \mathcal{N}\left(y; \sum_{i=1}^{N} h_i K_{\mathrm{m}}(\mu_i, \sigma_i), \sigma_\varepsilon\right),$$

where $\mathcal{N}(y; a, b)$ denotes the Gaussian distribution for the $y$ variable, with mean $a$ and standard deviation $b$. We can finally derive the *posterior* distribution as given by the Bayes theorem:

$$p(x|y) = \frac{p(y|x)p(x)}{p(y)}, \tag{11}$$

where $p(y)$ is a normalization constant whose value is unknown but can be neglected as it does not depend on $x$. The posterior distribution represents all the available information of the values of the unknown set of parameters $x$, given the data $y$. As such, it represents the solution to the inverse problem. However, it is a probability distribution on a relatively high-dimensional space $\mathcal{R}^{3N}$ and is therefore difficult to visualize. In the next section we illustrate a computational algorithm that allows samples from this distribution to be obtained and, in particular, the set of parameters that maximizes the posterior itself to be found.

Finally, let us observe that in this work we assume that $N$ is known; while this is, in general, not true, it makes the problem more tractable. In practice, $N$ can be selected by the user, and solutions with different values can be calculated and compared. Future work will be devoted to considering the general case of unknown $N$.

## 2.4 Monte Carlo algorithm

The posterior distribution defined in Eq. (11) is a complicated function on a relatively high-dimensional space: characterization of such distribution requires a computational tool that is able to deal with narrow peaks and local modes. In this work we produce a numerical approximation of the posterior distribution using a Markov chain Monte Carlo (MCMC) approach.

The general idea of Monte Carlo methods is to *sample* the target distribution $p(x|y)$, i.e., to obtain a set of realizations $x^{(1)}, \dots, x^{(N)}$ of random variables $X^{(1)}, \dots, X^{(n)}$ distributed according to the target distribution $p(x|y)$; indeed, the law of large numbers then guarantees that, for any suitable function $f(X)$, the following approximation holds:

$$\int_\Omega f(x)\pi(x|y)\mathrm{d}x \simeq \frac{1}{N}\sum_{i=1}^{N} f(x^{(i)}),$$

where $\Omega$ is a given domain. The possibility of approximating the integral of any function $f()$ implies that one has a full characterization of the target distribution.

MCMC methods work by constructing a Markov Chain whose invariant distribution is the target posterior distribution, i.e. the transition kernel $P(x'|x)$ of the Markov Chain satisfies

$$\pi(x'|y) = \int P(x'|x)\pi(x|y)\mathrm{d}x. \tag{12}$$

The implications of this choice are obvious: if $X^{(k)}$ is distributed according to $\pi(x|y)$, then each $X^{(n)}$ has the same distribution for $n \geq k$. Importantly, if a set of technical conditions are met, after an initial "wandering" time, referred to as *burn-in*, the Markov Chain will hit the target distribution *independently of the initialization*. Therefore, once a transition kernel satisfying Eq. (12) has been constructed, it is enough to simulate from it, starting with a random initialization.

In this work we use the Metropolis–Hastings construction of the kernel, which has the form

$$P(x'|x) = \alpha(x, x')Q(x'|x) + (1 - \alpha(x, x'))\delta(x', x) \tag{13}$$

and has to be interpreted as follows: given the current state $x$, the Markov Chain will evolve with probability $\alpha(x, x')$ according to a *proposal* distribution $Q(x'|x)$, and with probability $1 - \alpha(x, x')$ it will remain in $x$; the delta function $\delta(x', x)$ here is the Dirac delta that gives all the probability mass to the point $x' = x$. A sufficient condition that guarantees that the transition kernel defined in Eq. (13) is invariant with respect to $\pi(x|y)$ is that the *acceptance* probability

$\alpha(x, x')$ is set as follows:

$$\alpha(x, x') = \frac{\pi(x'|y) Q(x|x')}{\pi(x|y) Q(x'|x)}. \tag{14}$$

In the numerical simulations below we use Gaussian proposal distributions; Gaussian distributions have the pleasant property $Q(x'|x) = Q(x|x')$ so that the acceptance ratio simplifies, as detailed below.

Eventually, the MCMC algorithm works as follows. We start from an initializing value $x^{(1)}$, drawn from the prior distribution. Then for $k > 1$,

– for every and each parameter within $x^{(k)}$, a new value is proposed by drawing a Gaussian perturbation around the current value; this way, a new state is proposed $x^*$ which is identical to $x^{(k)}$ for all but one value (e.g., assuming $N = 1$, $x^* = (h^{(k)}, \mu^*, \sigma^{(k)})$, where $\mu^*$ is drawn from a Gaussian distribution of mean $\mu^{(k)}$).

– compute the acceptance ratio $\alpha$; given our previous choices of uniform distribution and symmetric proposal distribution,

$$\alpha = \frac{p(y|x^*)}{p(y|x^{(k)})}. \tag{15}$$

– accept the proposed value with probability $\min(1, \alpha)$: if $\alpha \geq 1$, then set $x^{(k+1)} = x^*$; otherwise, draw a uniform number $r \in [0, 1]$, and accept the proposed value if $r < \alpha$; otherwise, set $x^{(k+1)} = x^{(k)}$ (i.e., do not move).

The algorithm proceeds for a fairly large number of samples; in the simulations below, we use 5000 samples.

## 3 Numerical tests

We show several examples of applications of the Monte Carlo method to completely synthetic data as well as to data derived from experimental recordings, in the following denoted as *quasi-real* data. We first consider unimodal distributions with variable modal radius from "fine" to "coarse" and then with bimodal distributions with components of variable amplitude. Synthetic distributions are generated so as to represent typical atmospheric aerosol distributions; for this reason, an analysis of the distributions deriving from the observations of the AERONET network has been made. We then analyze three quasi-real datasets derived from the AERONET network. As already mentioned, throughout this study the complex refractive index $m$ is considered to be known; the possibility of also estimating the CRI will be the subject of future developments.

### 3.1 Performance evaluation

In order to perform a quantitative analysis of reconstruction accuracy, we use two different methods.

The first method is based on the deviation between the size distribution (SD) reconstructed by the inversion algorithm and the simulated exact SD. Indeed, the algorithm determines the SD that reproduces the set of measured parameters with some tolerance to account for the presence of noise; it is first necessary to define a method for quantitatively measuring the distance between the synthetic SD and the reconstructed SD. This can be done using the deviation defined as

$$\text{deviation} = \sum_{r_i = r_{\min}}^{r_{\max}} \left( \frac{\text{SD}_\text{T}(r_i) - \text{SD}_\text{R}(r_i)}{\sum_{r_k = r_{\min}}^{r_{\max}} \text{SD}_\text{T}(r_k) \frac{\Delta r_k}{r_k}} \right)^2 \frac{\Delta r_i}{r_i},$$

where TS2 $\text{SD}_\text{T}$ is the true size distribution (simulated); $\text{SD}_\text{R}$ is the size distribution reconstructed by the algorithm; and $r_{\min}$ and $r_{\max}$ are minimum and maximum values of the radius, determined on the basis of the condition that within the interval the $\text{SD}_\text{T}$ values are higher than $10^{-5} \times$ maximum of the $\text{SD}_\text{T}$. This last condition allows only the significant parts of the distribution to be taken into account.

The second method of evaluating the accuracy of the solutions is based on the calculation of integral properties of the size distributions. Since our algorithm allows us to determine the dimensional distribution expressed as

$$\frac{\text{d}N(r)}{\text{d}\ln(r)},$$

it is possible to derive the concentration in volume

$$\frac{\text{d}V(r)}{\text{d}\ln(r)} = \frac{4\pi r^3}{3} \frac{\text{d}N(r)}{\text{d}\ln(r)},$$

and therefore the particulate matter parameters $PM_1$, $PM_{2.5}$ and $PM_{10}$, defined as the amount of particulate matter with sizes smaller than 1, 2.5 and 10 μm, respectively, can be computed as

$$\text{PM} = \rho \int_0^R \frac{\text{d}N(r)}{\text{d}\ln(r)} \text{d}\ln(r),$$

where $\rho$ represents the particulate matter density, which is here assumed constant and equal to 1, and $R$ assumes the values 1, 2.5 or 10 μm for $PM_1$, $PM_{2.5}$ and $PM_{10}$, respectively.

In addition, we also take into consideration the effective radius $r_\text{eff}$ and the mean volume radius $r_\text{v}$, defined as

$$r_\text{eff} = \frac{\int_{r_{\min}}^{r_{\max}} r^3 \frac{\text{d}N(r)}{\text{d}\ln(r)} \text{d}\ln(r)}{\int_{r_{\min}}^{r_{\max}} r^2 \frac{\text{d}N(r)}{\text{d}\ln(r)} \text{d}\ln(r)}$$

and

$$\ln r_\text{v} = \frac{\int_{r_{\min}}^{r_{\max}} \ln(r) \frac{\text{d}N(r)}{\text{d}\ln(r)} \text{d}\ln(r)}{\int_{r_{\min}}^{r_{\max}} \frac{\text{d}N(r)}{\text{d}\ln(r)} \text{d}\ln(r)},$$

where $r_{\min}$ and $r_{\max}$ assume in our case the values 0.01 and 20 μm, respectively.

## 3.2    Numerical validation

We first proceed to show that the deviation measure is a good indicator of the performances in the sense that its value gives a quantitative evaluation of the distance between true and estimated size distributions. To this aim, we simulated in Simulation 1 a unimodal distribution with the parameters given in Table 1 (middle column). We then run the reconstruction algorithm multiple times, from different random initializations and with different numbers of iterations ranging between 100 and 5000, so as to reach a diverse population of final estimates. In the inversion we assumed that the optical input parameters $(2\alpha + 3\beta)$ are determined with the 5 % error and imposed the constraints on the parameters given in Table 1. We observe that a 5 % error in the retrieved optical parameters might seem unrealistically small; however, as the inverse method is applied to a single set of optical parameters, these can be obtained by averaging across different altitudes and/or times, thus effectively reducing the impact of noise and making our assumption plausible. Future studies will be devoted to investigate the impact of the noise level more in detail.

Discretization of the $r$ variable used 500 values logarithmically spaced between 0.01 and 20 μm. Notably, these values correspond to $r_a$ and $r_b$ in Eqs. (2) and (3); i.e., they define the discretization range in the integration and shall not be interpreted as extrema of a range for which the method can obtain reliable reconstructions. Indeed, it is typically hard to retrieve modes corresponding to particles larger than 7 μm.

Figure 1 shows the comparison between the simulated size distribution and the reconstructed size distribution in cases where the deviation is $< 0.0001$ (left), 0.001 (center) and 0.01 (right), respectively. This figure suggests that the deviation is a good indicator of the quality of the reconstruction.

Having ascertained that the deviation is a good measure of the "closeness" of the reconstruction to the real distribution, it is obvious to assume that a reconstruction with a small value of deviation must correspond to a low value of the *discrepancy* between measured and predicted optical coefficients, but the inverse statement is not true. In fact, since the discrepancy only measures the distance of the reconstructed optical parameters with respect to the input ones, small differences in the optical parameters (low values of the discrepancy) can correspond to very large differences in the size distribution parameters.

The statistical nature of the Monte Carlo method includes an intrinsic instability in the sense that a repetition of the calculation with the same set of input optical parameters, even if without error, leads to different reconstructions. The dispersion of the reconstructions with the same initial conditions is a measure of the stability of the method.

Another issue is the effect of noise on the optical input parameters. This random perturbation causes a further increase in the instability of the reconstruction, which may also prevail over the intrinsic instability of the method.

In order to have a quantitative evaluation of the influence of the instability of the method with respect to the noise in the input parameters, we have made a statistical analysis of the discrepancy. A bimodal distribution was simulated with the parameter values given in Table 1

The values of the extinction coefficients at the wavelengths of 355 and 532 nm and of the backscattering coefficients at the wavelengths of 355, 532 and 1064 nm were then determined with the Mie theory, considering homogeneous spherical particles. The values of the optical coefficients were then used as input data for the reconstruction. The reconstruction was repeated 30 times, each time perturbing the set of input parameters in order to simulate a 5 % error in each of them. We then calculated the discrepancy of each of the 30 sets of optical parameters perturbed with respect to the unperturbed set (input discrepancy). The distribution of the input discrepancy is shown in Fig. 2, left plot, and the distribution of the values of the discrepancy of the corresponding 30 reconstructions (output discrepancy) is shown in the right plot in the same figure.

It should be kept in mind that the discrepancy input represents the "distance" between each set of perturbed parameters and the theoretical set, while the discrepancy output represents the "distance" between the set of parameters corresponding to each reconstruction with respect to the set of input parameters of the same reconstruction. From Fig. 2 we see that (1) the uncertainty of 5 % in the optical input parameters corresponds to a much greater fluctuation in the solution than the one produced by the instability of the algorithm, and (2) the algorithm always finds the solutions corresponding to a set of optical parameters very close to the input one. To take this into account, our method is based on repetition of the calculation; at each repetition, the set of optical parameters is disrupted, and each of the parameters is subjected to a variation that takes into account the experimental error. In practice, each parameter is assigned a value extracted from a Gaussian distribution around the experimental value and width equal to the error itself. With each set of optical input parameters thus determined, the Monte Carlo is executed with a fixed number of iterations. The standard deviation of the coefficients of the size distributions thus obtained is considered a good evaluation of the uncertainty in the final solution.

The uncertainty obviously takes into account both the instability of the method and the errors in the input parameters. In the tests conducted so far the distribution of the optical parameters has been considered Gaussian, but in view of the simple logical structure of the algorithm, it is in principle possible to introduce arbitrary distributions to take into account, for example, the contribution of systematic errors in the input optical parameters (consider for example that the error in the backscattering coefficient at 1064 nm can be dominated by the uncertainty in the lidar ratio, whose value should be fixed a priori in a more or less arbitrary manner).

**Table 1.** True values of log-normal distributions used for testing purposes in Sect. 3.2 and corresponding constraints imposed in the inversion algorithm.

| Parameter | Simulation 1 Mode | Simulation 2 Mode 1 | Mode 2 | Inversion constraints |
|---|---|---|---|---|
| $h$ | 0.02 | 0.02 | 0.02 | 0.0001–0.3 |
| $\sigma$ | 1.52 | 1.52 | 2.4 | 1.3–5.5 |
| $\mu$ [μm] | 1.56 | 0.15 | 1.35 | 0.1–20 |
| $Re$ | 1.49 | 1.49 | 1.49 | 1.49 |
| Im | 0.019 | 0.019 | 0.019 | 0.019 |

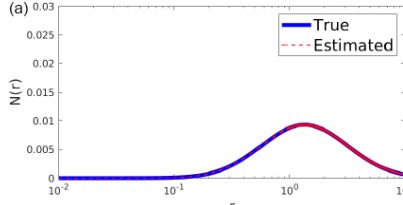 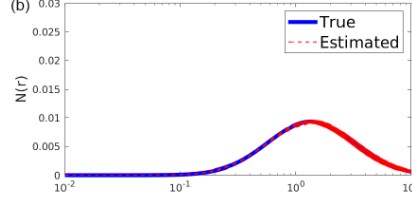 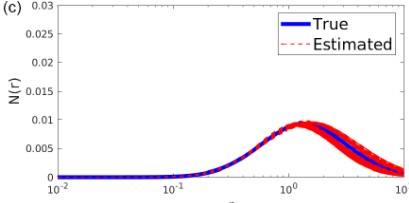

**Figure 1.** Reconstructions of number size distribution that show increasingly large deviation from the true one; from **(a)** to **(c)**, we show reconstructions whose deviation is lower than 0.0001 **(a)**, 0.001 **(b)** and 0.01 **(c)**.

### 3.3 Results with synthetic data

In the following we show the results of different tests for simulated SDs representing realistic cases. The reconstruction has been obtained by setting the number of iterations to 5000 and by running the algorithm 30 times, with noise equal to 5 %. For each run, we collect the best solution, and we provide uncertainty quantification, shown as a shaded area in the pictures below, using the standard deviation of the best solution across these runs.

Figure 3 shows the reconstructions of unimodal distributions obtained by requiring the reconstruction to be unimodal and by using the parameters listed in Table 2.

The (a) and (b) cases simulate a unimodal SD centered on the coarse mode of a realistic bimodal distribution; the (c) and (d) cases simulate a unimodal SD which approximates a fine mode of a realistic bimodal distribution.

In Table 3 we report the deviation for each of the reconstructions in Fig. 3 and the percentage deviation of the parameters $PM_1$, $PM_{2.5}$, $PM_{10}$, PM, $r_{eff}$ and $r_v$ with respect to the simulated SD.

Figure 4 shows the reconstructions with minimum discrepancy of a simulated bimodal distribution with two modes with similar width. The goal of this test is to evaluate the accuracy of the reconstruction of a realistic bimodal distribution obtained by a unimodal distribution whose parameters have the same constraint as those in Fig. 3.

Figures 5 and 6 show the reconstructions of the same distributions simulated in Figs. 3 and 4, obtained by using a bimodal distribution with the parameters given in Table 2.

**Table 2.** True values of unimodal and bimodal log-normal distributions used for testing purposes in Sect. 3.3 and corresponding constraints imposed in the inversion algorithm.

| Parameter | Unimodal case (a) | (b) | (c) | (d) | Inversion constraints |
|---|---|---|---|---|---|
| $h$ | 0.025 | 0.05 | 0.02 | 0.02 | 0.0002–0.3 |
| $\sigma$ | 2 | 2 | 1.7 | 1.7 | 1.3–5.0 |
| $\mu$ [μm] | 2.5 | 2.5 | 0.5 | 0.2 | 0.1–20 |
| | Bimodal case | | | | |
| $h_1$ | 2 | 2 | 0.02 | 0.02 | 0.00005–0.2 |
| $\sigma_1$ | 1.7 | 1.7 | 1.7 | 1.7 | 1.3–2.8 |
| $\mu_1$ [μm] | 0.5 | 0.5 | 0.2 | 0.2 | 0.1–0.9 |
| $h_2$ | 0.025 | 0.05 | 0.025 | 0.05 | 0.00005–0.3 |
| $\sigma_2$ | 2 | 2 | 2 | 2 | 1.3–2.8 |
| $\mu_2$ [μm] | 2.5 | 2.5 | 2.5 | 2 | 1–5.2 |
| $Re$ | 1.49 | 1.49 | 1.49 | 1.49 | 1.49 |
| Im | 0.019 | 0.019 | 0.019 | 0.019 | 0.019 |

Tables 3, 4, 5 and 6 report the deviations of each reconstruction of the Figs. 3, 4, 5 and 6 and the percentage deviations of the parameters $PM_1$, $PM_{2.5}$, $PM_{10}$, PM, $r_{eff}$ and $r_v$ with respect to the simulated SD.

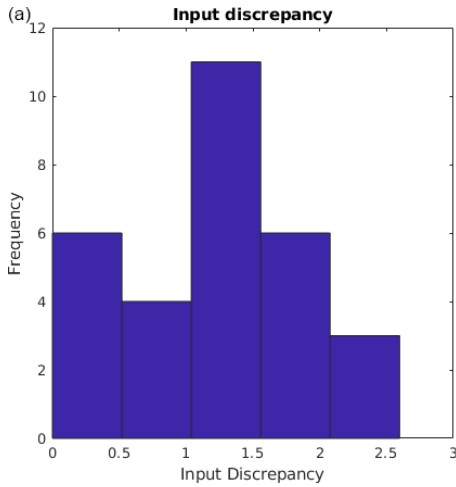
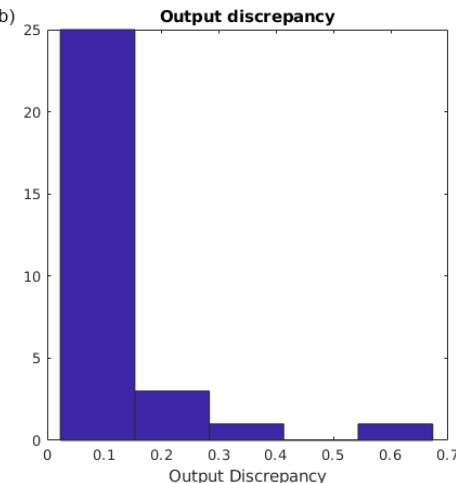

**Figure 2.** The algorithm always provides output (predicted) optical parameters $\alpha$ and $\beta$ very close to the true ones: over 30 repetitions, in **(a)** we plot a histogram of the discrepancies between the input noisy parameters and the exact ones; in **(b)**, we plot a histogram of the discrepancies between the output parameters and the exact ones. Notice also the different scale on the $x$ axis.

**Table 3.** Performance metrics obtained by running the inversion algorithm with a unimodal distribution, when the data are generated by a synthetic unimodal distribution (see Fig. 3).

|  | a | b | c | d |
|---|---|---|---|---|
| SD deviation ($10^{-3}$) | 1.5 | 0.1 | 0.0 | 10 |
| $\Delta$PM$_1$ (%) | 0.7 | 0.9 | −0.1 | −1.6 |
| $\Delta$PM$_{2.5}$ (%) | −0.6 | −1.1 | −0.05 | −1.6 |
| $\Delta$PM$_{10}$ (%) | 4.6 | −1.0 | −0.05 | −1.6 |
| $\Delta$PM (%) | −5.0 | −0.9 | −0.05 | −1.6 |
| $\Delta r_{\text{eff}}$ (%) | −3.0 | −0.3 | −0.2 | 11.4 |
| $\Delta r_{\text{v}}$ (%) | 1.6 | −2.7 | −0.7 | 25.6 |

**Table 5.** Performance metrics obtained by running the inversion algorithm with a bimodal distribution, when data are generated by a synthetic unimodal distribution (see Fig. 5).

|  | a | b | c | d |
|---|---|---|---|---|
| SD deviation ($10^{-3}$) | 2.5 | 15 | 0.08 | 24 |
| $\Delta$PM$_1$ (%) | −6.8 | −7.5 | −0.8 | 2.3 |
| $\Delta$PM$_{2.5}$ (%) | −1.9 | 0.8 | −0.6 | 6.2 |
| $\Delta$PM$_{10}$ (%) | 4.4 | 5.2 | −0.4 | 17.9 |
| $\Delta$PM (%) | 4.7 | 5.2 | −0.4 | 21.2 |
| $\Delta r_{\text{eff}}$ (%) | 2.8 | 3.9 | −0.7 | 4.5 |
| $\Delta r_{\text{v}}$ (%) | −8.7 | −6.5 | 2.1 | −21.5 |

**Table 4.** Performance metrics obtained by running the inversion algorithm with a unimodal distribution, when data are generated by a synthetic bimodal distribution (see Fig. 4).

|  | a | b | c | d |
|---|---|---|---|---|
| SD deviation ($10^{-3}$) | 54 | 46 | 120 | 67 |
| $\Delta$PM$_1$ (%) | 5.2 | 6.9 | 14.4 | 20.6 |
| $\Delta$PM$_{2.5}$ (%) | −5.34 | −3.9 | −12.5 | −5.6 |
| $\Delta$PM$_{10}$ (%) | −26 | −23 | −35.1 | −20.4 |
| $\Delta$PM (%) | −27 | −24 | −35.9 | −17.8 |
| $\Delta r_{\text{eff}}$ (%) | −32 | −26 | −34.8 | −46.9 |
| $\Delta r_{\text{v}}$ (%) | −56 | −64 | −65.9 | −85.2 |

**Table 6.** Performance metrics obtained by running the inversion algorithm with a bimodal distribution, when the data are generated by a synthetic bimodal distribution (see Fig. 6).

|  | a | b | c | d |
|---|---|---|---|---|
| SD deviation ($10^{-3}$) | 11.0 | 25 | 100 | 6 |
| $\Delta$PM$_1$ (%) | −0.4 | 4.1 | −1.9 | 3.8 |
| $\Delta$PM$_{2.5}$ (%) | 45.TS3 | −6.3 | 1.9 | −0.7 |
| $\Delta$PM$_{10}$ (%) | 10.9 | −18.4 | 34.5 | 5.9 |
| $\Delta$PM (%) | 10.4 | −18.4 | 43.9 | 7.7 |
| $\Delta r_{\text{eff}}$ (%) | 7.5 | −24.3 | 37.5 | −9.5 |
| $\Delta r_{\text{v}}$ (%) | 2.4 | −57.5 | −16.4 | −24.6 |

### 3.4 Results with quasi-real data

We finally validate our proposed method on three datasets that have been obtained from experimental data recorded by AERONET by applying the direct calculation of Mie functions to the size distribution reported in the AERONET database. For all three datasets, we attempt reconstruction with a unimodal and a bimodal distribution: constraints for the parameter values are reported in Table 7. In order to exemplify how the quality of the retrieval may depend on the actual value of the CRI, in addition to using the CRI value estimated by AERONET, we also use a very different value, i.e., one with a larger imaginary part. Specifically, the used CRI ($1.57 + i0.43$) corresponds to the maximum value of

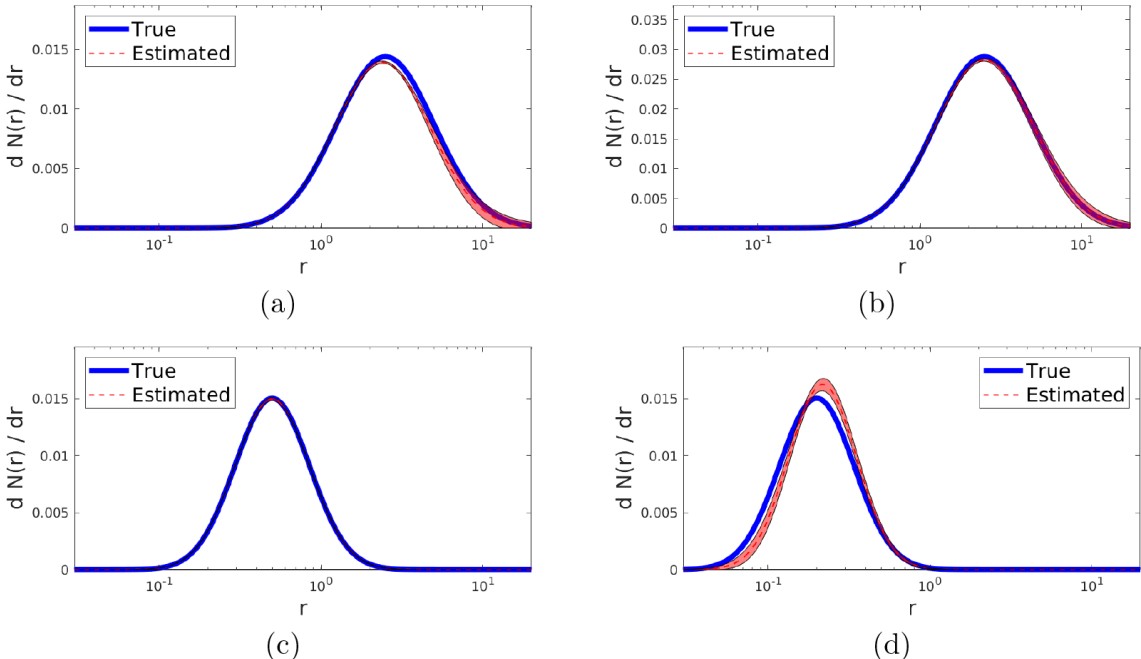

**Figure 3.** Reconstructions obtained by running the inversion algorithm with a unimodal distribution, when data are generated by a unimodal distribution. Different panels from **(a)** to **(d)** correspond to different parameters characterizing the unimodal distributions: the respective parameter values are given in Table 2 ("Unimodal case" section).

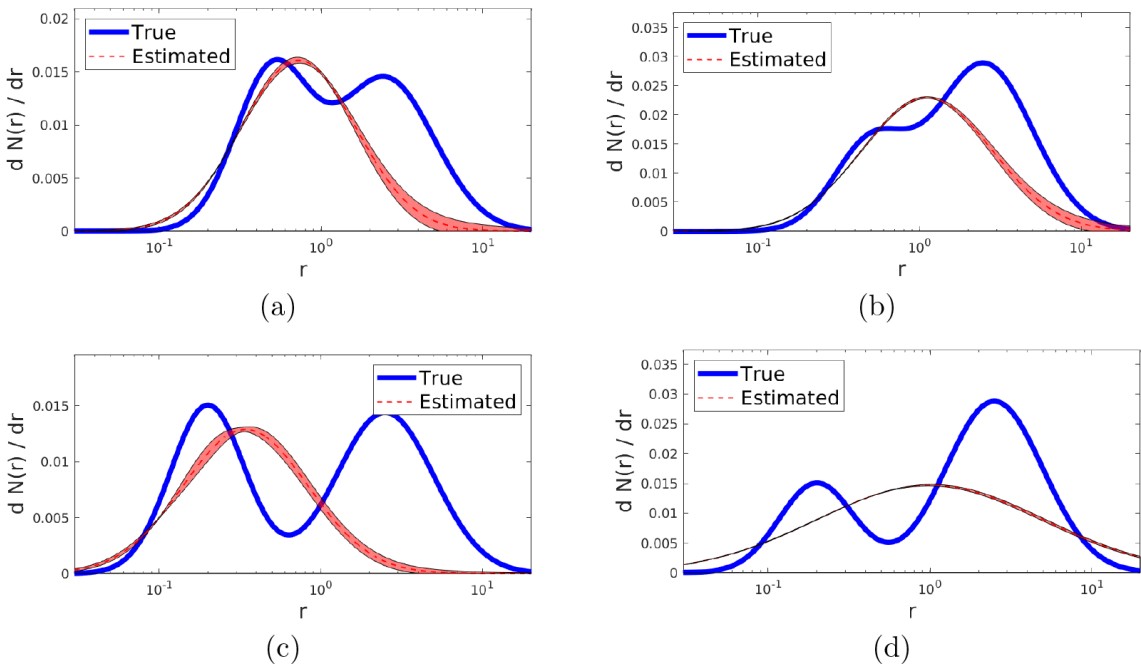

**Figure 4.** Reconstructions obtained by running the inversion algorithm with a unimodal distribution, when data are generated by a bimodal distribution. Different panels from **(a)** to **(d)** correspond to different parameters characterizing the bimodal distributions: the respective parameter values are given in Table 2 ("Bimodal case" section for the data generation, "Unimodal case" section for the inversion constraints).

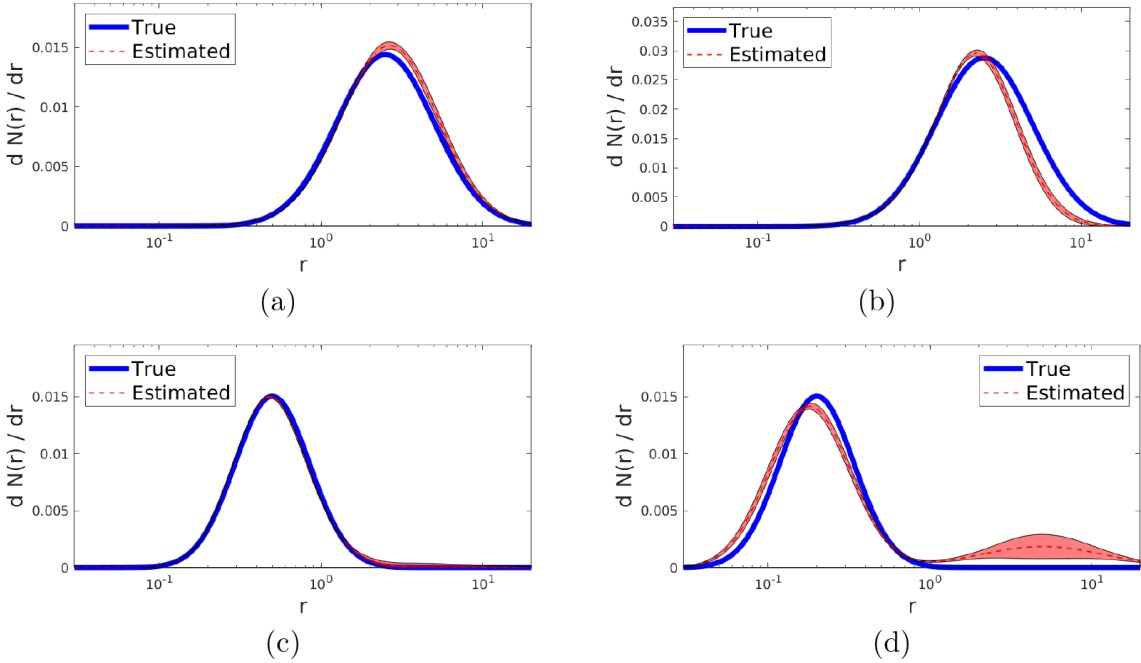

**Figure 5.** Reconstructions obtained by running the inversion algorithm with a bimodal distribution, when data are generated by a unimodal distribution. Different panels from **(a)** to **(d)** correspond to different parameters characterizing the unimodal distributions: the respective parameter values are given in Table 2 ("Unimodal case" section for the data generation, "Bimodal case" section for the inversion constraints).

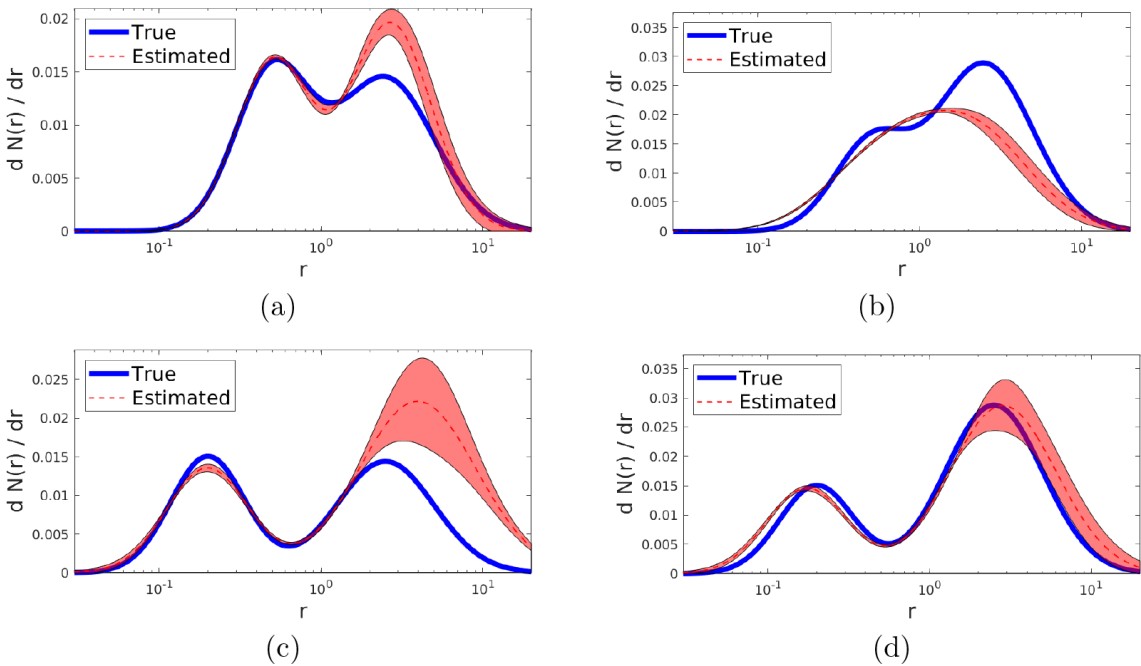

**Figure 6.** Reconstructions obtained by running the inversion algorithm with a bimodal distribution, when data are generated by a bimodal distribution. Different panels from **(a)** to **(d)** correspond to different parameters characterizing the bimodal distributions: the respective parameter values are given in Table 2 ("Bimodal case" section).

**Table 7.** Constraints used in the analysis of the quasi-real data.

| Parameter | Unimodal case 1 Mode (min–max) | Bimodal case Mode 1 (min–max) | Mode 2 (min–max) |
|---|---|---|---|
| $h$ | 0.0001–0.25 | 0.0001–0.11 | 0.0001–0.25 |
| $\log \sigma$ | 1.35–4 | 1.35–4 | 1.35–4 |
| $\mu$ [µm] | 0.06–5.7 | 0.06–0.5 | 0.4–5.7 |

**Table 8.** Details of the three experimental datasets used in Sect. 3.4, algorithm settings and the corresponding performance evaluation.

| Site | Date | Modes | Refractive index | $\Delta PM_1$ (%) | $\Delta\,PM_{2.5}$ (%) | $\Delta\,PM_{10}$ (%) | $\Delta PM_{10\,TOT}$ (%) |
|---|---|---|---|---|---|---|---|
| Bucharest Inoe | 22 July 2010 | 1 | $1.37 + i0.0057$ | 23.5 | 8.9 | −8.8 | −9.7 |
| | | 2 | $1.37 + i0.0057$ | 8.8 | 8.9 | 16.2 | 18.2 |
| | | 2 | $1.57 + i0.043$ | −2.4 | 4.2 | 60 | 73 |
| Etna | 17 July 2016 | 1 | $1.57 + i0.0017$ | 29 | −23 | −25 | −24.9 |
| | | 2 | $1.57 + i0.0017$ | 0.3 | 0.6 | −14.9 | −15.1 |
| | | 2 | $1.57 + i0.043$ | 14.9 | −15.7 | −22.9 | −22.9 |
| Gozo | 2 January 2015 | 1 | $1.41 + i0.0005$ | 24 | 9.3 | −16 | −14.8 |
| | | 2 | $1.41 + i0.0005$ | −2.0 | 15 | −0.7 | −1.4 |
| | | 2 | $1.57 + i0.043$ | −1.5 | 2.3 | 39.8 | 54.4 |

both the real and the imaginary part as measured for the Saharan dust in several measurement campaigns (Wagner et al., 2012).

All reconstructions were obtained with 5000 iterations, 20 repetitions and 5 % noise in the optical parameters. In Table 8 we report details of the datasets, including registration site and date as well as performance evaluation of our reconstructions.

## 4 Discussion

The comparisons between the reconstructed and the simulated distributions, shown in Figs. 3–6, with 5 % noise in the optical parameters allow some conclusions to be drawn.

As shown in Fig. 3 and Table 3, the reconstructions of simulated unimodal distributions with unimodal distributions give excellent results. The distributions with modal radius between 0.2 and 2.5 µm are reconstructed with high accuracy both for the shape of the distribution and for the integral properties ($PM_1$, $PM_{2.5}$, $PM_{10}$, PM). More critical is the reconstruction of the effective radius and the mean volume radius; however the worst case happens when the size distribution is simulated with modal radius equal to 0.2 mm, in which case the difference with the values of the simulated distribution is equal to 11 % and 25 % for $r_{\rm eff}$ and $r_{\rm v}$, respectively. Note that the above results are obtained with very large, not realistic, constraints to the parameters of the reconstructed distribution.

The reconstruction of bimodal distributions, with realistic values of the parameters, by using unimodal distributions (see Fig. 4 and Table 4) gives solutions which show very important deviations with respect to the simulated distributions; nevertheless, the values of the integral parameters have deviations which are less than 35 %. In these cases, $r_{\rm eff}$ and $r_{\rm v}$ are always underestimated and show percentage deviations with respect to the real value of 50 % and 80 %, respectively. These values are comparable to those routinely obtained in the inversion of lidar data, which typically provide errors around 50 % (Di et al., 2018a).

Figures 5 and 6 and the respective Tables 5 and 6 refer to reconstructions with bimodal SDs of a simulated unimodal distribution and a simulated bimodal distribution, respectively. In this case, the reconstructions are done by imposing more strict constraints than in the cases of unimodal distributions. The constraints are the product of a statistical analysis of the parameters of the size distributions from the AERONET network. In these cases the reconstruction of unimodal distributions gives solutions which have deviation less than 5 % except for the case of distributions with modal radius equal to 0.2 µm, in which case $PM_{10}$ and PM have deviations around 8 % and 21 %, respectively; such deviations (see Fig. 5d) are due to the fact that often the algorithm introduces some contribution of particles with a radius around 5 µm. In all the cases, $r_{\rm eff}$ is reconstructed with an accuracy of around 5 %, while the accuracy of $r_{\rm v}$ is less than 9 %, but in the case of distribution simulated with modal radius equal to 0.2 mm, $r_{\rm v}$ is underestimated by 22 %.

In the bimodal SD reconstruction (Fig. 6 and Table 6), the "fine" mode is always reconstructed better than the "coarse" mode. This effect is connected to the wavelengths used for determining the optical parameters of the lidar measures.

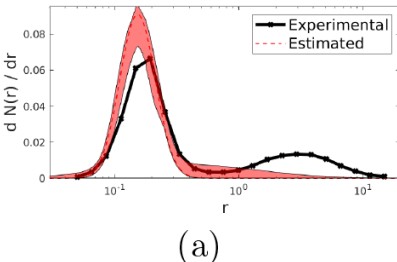 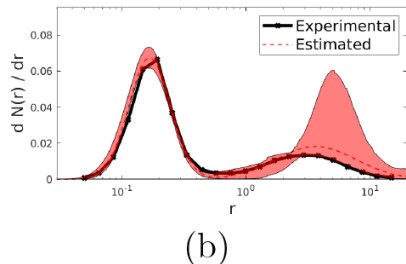 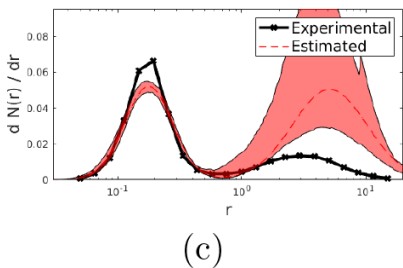

**Figure 7.** Results obtained by applying our proposed method to the experimental dataset corresponding to the AERONET size distribution measured in Bucharest: in **(a)** and **(b)** we show the retrieval with a unimodal and bimodal distribution, respectively, using the CRI determined by AERONET. In **(c)** we show, as a comparison, the retrieval with a bimodal distribution using an extreme CRI value of $1.57 + i0.043$.

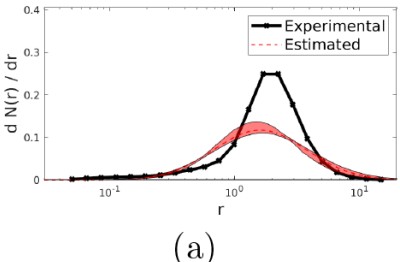 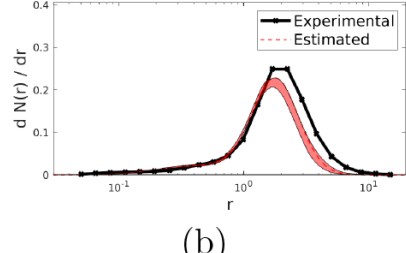 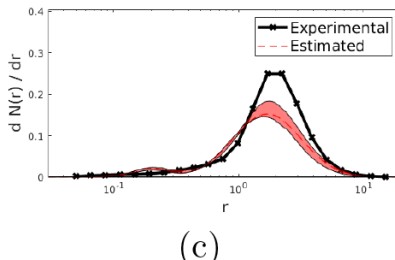

**Figure 8.** Results obtained by applying our proposed method to the experimental dataset corresponding to the AERONET size distribution measured on Etna: in **(a)** and **(b)** we show the retrieval with a unimodal and bimodal distribution, respectively, using the CRI determined by AERONET. In **(c)** we show, as a comparison, the retrieval with a bimodal distribution using an extreme CRI value of $1.57 + i0.043$.

Overall the integral properties are reconstructed with an accuracy of less than 6 % for $PM_1$ and $PM_{2.5}$ and an accuracy between 6 % and 44 % for $PM_{10}$ and PM; $r_{eff}$ and $r_v$ are recovered with an accuracy between 7 % and 37 % and between 2 % and 57 %, respectively.

The analysis of quasi-real data (Figs. 7–9) confirmed that the inversion algorithm is capable of recovering realistically shaped SDs.

We observe that reconstructions obtained with bimodal distributions provide consistently better results than those obtained with unimodal distributions and that the error in assessing the PM concentrations remains at more-than-acceptable levels, particularly with bimodal distributions. This result is particularly reasonable because all the datasets were indeed generated by bimodal distributions, including the one recorded on Etna (Fig. 8), which may be erroneously thought to correspond to a unimodal distribution.

Our analysis also highlights a few limitations of the proposed technique. First, the technique presents an inherent subjectivity as regards the choice of unimodal versus bimodal distributions; while, on average, the bimodal settings performs better, it can also produce some spurious modes such as those in Fig. 5d. Our recommendation is to use a unimodal distribution when there is strong a priori indication in favor of it and a bimodal distribution elsewhere. Future studies will investigate the possibility of including the number of modes among the unknowns and provide a posterior probability for different numbers of modes in the same fashion as was done in Sorrentino et al. (2014) for a neuroimaging application.

A second limitation concerns the subjectivity in the choice of the CRI, which was assumed to be known in the present study. Our analysis of quasi-real data showed how the quality of the retrieval may depend on the value of the CRI and partly deteriorates when the imaginary part grows, particularly for larger modes. This is a known issue with lidar data that can possibly be solved in a Bayesian framework by devising better priors. In addition, a full Bayesian model including the CRI among the unknowns can be devised; however, with an increased number of unknowns it will be necessary to exploit more prior information to reduce the degree of ill-posedness.

To conclude, we observe that the uncertainty quantification currently implemented seems to provide, at times, optimistic results, to the extent that the true distribution is not always included in the confidence bands. We reckon that these limitations can be overcome by using more complex but more powerful Monte Carlo sampling techniques, such as those described in Sorrentino et al. (2014), Luria et al. (2019), Sciacchitano et al. (2019) and Viani et al. (2021); this will also be the topic of future studies.

## 5  Conclusions

The preliminary results presented in this paper indicate that the proposed method can retrieve uni- and bimodal distri-

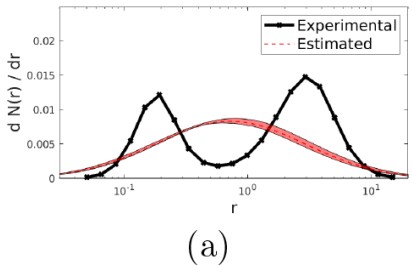
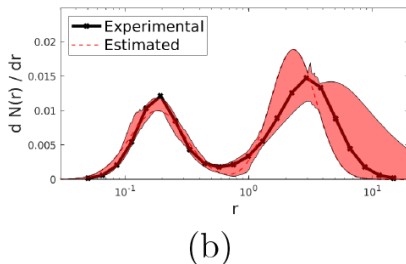
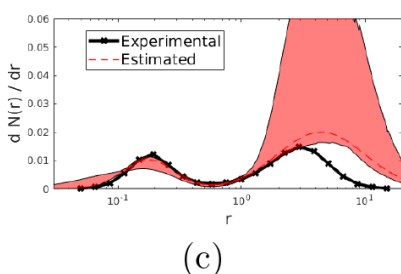

**Figure 9.** Results obtained by applying our proposed method to the experimental dataset corresponding to the AERONET size distribution measured in Gozo: in **(a)** and **(b)** we show the retrieval with a unimodal and bimodal distribution, respectively, using the CRI determined by AERONET. In **(c)** we show, as a comparison, the retrieval with a bimodal distribution using an extreme CRI value of $1.57 + i0.043$.

butions from extinction coefficients measured at two wavelengths and backscattering coefficients measured at three wavelengths when the correct number of modes is selected. The reconstruction of three-modal distributions is more challenging, and more constraints might be necessary to obtain reliable and stable solutions. The extension of the method to three-modal distributions and variable refractive index, together with better uncertainty quantification and automatic model selection, will be the subject of future studies.

*Code availability.* The code used in this article is not publicly accessible as the research has been partially funded by a private company. However, the article contains information that shall enable the reproduction of the code.

*Data availability.* Part of the data used in this article are publicly available on the AERONET website (https://aeronet.gsfc.nasa.gov/, NASA, 2021 TS4); the other ones are available on request, by sending an email to the authors.

*Author contributions.* ASo conceived and implemented the algorithm. VT and PC contributed to an efficient implementation of the algorithm. NS, ASa and AB performed the statistical analysis used to determine inversion constraints. NS ran the simulations and performed the analysis of experimental data. All authors contributed to writing the manuscript.

*Competing interests.* The contact author has declared that neither they nor their co-authors have any competing interests.

*Financial support.* Alberto Sorrentino was partially supported by Gruppo Nazionale per il Calcolo Scientifico. This project was partially supported by the Beijing Research Institute on Telemetry (BRIT), Beijing, China.

*Review statement.* This paper was edited by Daniel Perez-Ramirez and reviewed by three anonymous referees.

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

## Remarks from the typesetter