# Peer review of "A Bayesian parametric approach to the retrieval of the atmospheric number size distribution from lidar data"

_Atmospheric Measurement Techniques, 2021_

## Referee Comment (RC1)

Authors use Bayesian approach and Monte Carlo algorithm to retrieve the particle size distribution from so called 3+2 observations. This is definitely important task, but authors in beginning of manuscript, should clearly formulate: how many unknowns they have (six at least) and how many input data they use (five). The problem is underdetermined and unique solution does not exist. So authors should clearly explain how do they treat this issue. The plans to retrieve three – modal distributions in the future look unrealistic.

5% error in input data is rather optimistic assumption, it can be up to 10% or even higher. How level of errors in input data will influence the retrieval? Authors should clearly formulate what is advantage of their method comparing to existing approaches.

Authors assume that refractive index is knows. This is a weak point of this research. I think that variation of RI will strongly complicate the retrieval. And application of algorithm to real lidar measurements is very desirable.

Technical comments
Ln 11-14. Can be skipped. It is well known.

Ln 15-19. There are too many papers on this subject. Better refer to books or reviews.

Ln 27. In order to retrieve the microphysical properties of the aerosol from lidar measurements, two inverse problems must be solved in sequence: in the first inverse problem, one uses the measured backscattered power to obtain an estimate of the aerosol optical parameters
I am not sure that calculation of backscattering and extinction coefficients should be called an inverse problem.

Ln.31 estimate of the number size distribution
Why do author consider number size distribution? Normally volume distribution is used. Number distribution can be tricky when fine particles are considered.

Ln 114. total number of parameters to be estimated is therefore 3N, which is substantially smaller than the number of parameters
So for bimodal PSD you have 6 unknown parameters plus unknown real and imaginary part. How many input data do you need?

Fig.3-6 In caption should be given parameters used in calculations.

---

## Author Response (AR1)

**Response letter to the reviewers' comments on manuscript**
**"Retrieving the atmospheric number size distribution from lidar data"**

In the following, the reviewers' comments are reported in italic, and the Authors' responses are marked with an "A." and given in plain text.

**Reviewer 1**

*Authors use Bayesian approach and Monte Carlo algorithm to retrieve the particle size distribution from so called 3+2 observations. This is definitely important task, but authors in beginning of manuscript, should clearly formulate: how many unknowns they have (six at least) and how many input data they use (five). The problem is underdetermined and unique solution does not exist. So authors should clearly explain how do they treat this issue. The plans to retrieve three – modal distributions in the future look unrealistic.*

A.  We thank the reviewer for this comment; we added two sentences in the Introduction that specify better the problem in numerical terms, as well as the Bayesian approach and its advantages:

"the number of unknowns to be estimated is either three, in the unimodal case, or six, in the bimodal case; the problem is therefore over-determined in the unimodal case and under-determined in the bimodal case. In both cases, it is very useful to have the possibility to exploit any information one might have a priori, for instance on the plausible interval range for the values of these unknowns; to this aim, we set up a Bayesian model where such a priori information can be coded in proper prior distributions, and we then use uniform priors in selected intervals"

*5% error in input data is rather optimistic assumption, it can be up to 10% or even higher. How level of errors in input data will influence the retrieval?*

A.  We thank the reviewer for this comment: we added the following sentence to better address this point

"We observe that a 5% error on the retrieved optical parameters might seem unrealistically small; however, as the inverse method is applied to a single set of optical parameters, these can be obtained by averaging across different altitudes and/or times, thus effectively reducing the impact of noise and making our

assumption plausible. Future studies will be devoted to investigate the impact of the noise level more in detail."

*Authors should clearly formulate what is advantage of their method comparing to existing approaches.*

A. We thank the reviewer for their comment. We added the following text in the Introduction, where we explain the advantages of the proposed method:

"We reckon the proposed method features three main advantages with respect to the state-of-the-art: (i) because it is based on a Bayesian model, it naturally provides uncertainty quantification on the estimated parameters, which is not always the case for competitors; (ii) because it makes use of a Monte Carlo algorithm, it does not get stuck in local minima like deterministic optimization algorithms often do, and (iii) for the same reason it can be easily generalized to include, for instance, a non-Gaussian distribution of the noise term."

*Authors assume that refractive index is knows. This is a weak point of this research. I think that variation of RI will strongly complicate the retrieval. And application of algorithm to real lidar measurements is very desirable.*

A. We thank the reviewer for this comment. Indeed, the variation of RI does complicate the retrieval and additional priors/constraints are needed to provide results; this is something we plan to work on in the near future.
As far as application to real lidar measurements is concerned, we agree that this is important, however, to the best of our knowledge there are no lidar data where the "true solution" is known (i.e. simultaneous colocated measurements that can confirm the goodness of the solution obtained from lidar data); since this article is a methodological one, introducing the method, we felt it was necessary to always provide the true underlying solution together with the retrieval. In future studies we will be using the method with experimental data.

Technical comments

*Ln 11-14. Can be skipped. It is well known.*

A. We agree with the reviewer, however, we preferred to leave those three lines as introductory lines for potential readers coming from other disciplines.

*Ln 15-19. There are too many papers on this subject. Better refer to books or reviews.*

A. We thank the reviewer for their comment; unfortunately we are not aware of recent review articles on this topic: if the reviewer has a specific suggestion, we would be happy to add it to the cited References. We currently added reference to a book [Weitkamp, 2006] that contains a review chapter, however, the book is relatively old and outdated.

*Ln 27. In order to retrieve the microphysical properties of the aerosol from lidar measurements, two inverse problems must be solved in sequence: in the first inverse problem, one uses the measured backscattered power to obtain an estimate of the aerosol optical parameters*
*I am not sure that calculation of backscattering and extinction coefficients should be called an inverse problem.*

A. We think it is; A.S. actually published a paper on this topic in the journal *Inverse Problems* (Denevi et al. 2017). The term inverse problem is relatively ambiguous but mostly used to denote problems where one goes "backward" from measured data to unknown causes, through some known physical model. This is exactly the case with the calculation of backscattering and extinction coefficients.

*Ln.31 estimate of the number size distribution*
*Why do author consider number size distribution? Normally volume distribution is used.*
*Number distribution can be tricky when fine particles are considered.*

A. We thank the reviewer for this comment. Since there is a one-to-one correspondence between number size distributions and volume distributions, one can reconstruct the number size distribution and then transform it later to a volume distribution. It is true that some retrieval methods may provide different results when using different parametrizations, but this is mostly due to numerical reasons related to inversion of matrices and should not affect our Bayesian method with MCMC algorithm.

*Ln 114. total number of parameters to be estimated is therefore 3N, which is substantially smaller than the number of parameters*
*So for bimodal PSD you have 6 unknown parameters plus unknown real and imaginary part.*
*How many input data do you need?*

A. In fact, in this study we are assuming the Refractive Index to be known, which implies only 6 parameters need to be recovered from the 5 data, for bimodal distributions.

*Fig.3-6 In caption should be given parameters used in calculations.*

A. Thanks for this comment: we added a reference to Table 2 where the values are shown.

**Reviewer 2**

*The Manuscript is interesting to read and provides a new idea in using a Bayesian model and a Monte Carlo algorithm under a few strong assumptions:*

*(1) The complex refractive index (CRI) is fixed (wavelength independent) and must be known a priori. Therefore, the title of the manuscript is not appropriate, since lidar data do not provide the CRI.*

    A. We thank the reviewer for this comment; we modified the title, that now reads "A Bayesian parametric approach to the retrieval of the atmospheric number size distribution from lidar data"

*(2) The modes (fine, coarse...) are fixed log-normal distributions.*

    A. This is true. In general in inverse problems one is almost systematically faced with a choice of this type: (a) using a simple parametric model is not always correct but provides simple, interpretable results and (when combined with a Bayesian model) uncertainty quantification that often helps to assess whether the model is working fine; or (b) using a complex, nonparametric model is more general but makes the result more complex to interpret and it is usually difficult to provide uncertainty quantification. In our view both choices can be equally criticized.

*(3) The number of modes has to be known a priori, otherwise the retrieval could fail, see given examples.*

    A. True. We are working on a generalization where the number of modes will be estimated from the data, but in its current form one has to choose subjectively. We added a recommendation in the manuscript: "Our recommendation is to use a unimodal distribution when there is strong \textit{a priori} indication in favor of it, and a bimodal distribution elsewhere."

*(4) The values r_min and r_max are essential values, too, known from other References. In line 203 one learns that these values also must be known a priori.*

    A. We thank the reviewer for this comment. We now specify much earlier, in the Introduction, that we use uniform prior distributions over selected intervals.

*(5) The tested error level of 5% is too small for lidar data.*

    A. We thank the reviewer for this comment: we added the following sentence to better address this point

    "We observe that a 5% error on the retrieved optical parameters might seem unrealistically small; however, as the inverse method is applied to a single set of optical parameters, these can be obtained by averaging across different altitudes and/or times, thus effectively reducing the impact of noise and making our

assumption plausible. Future studies will be devoted to investigate the impact of the noise level more in detail."

*Other remarks:*

*Line 137 misprint*

    A. Corrected.

*Line 154 misprint*

    A. Corrected

*Equation (13): What is \delta?*

    A. We now specify \delta is the Dirac delta

*What is the difference between r_a and r_min and r_b and r_max, respectively?*

    A. The difference is partly conceptual -- r_a and r_b pertain to the forward computation, while r_min and r_max are constraints on the inverse method; practically, we agree that their values should coincide when a unimodal distribution is considered; when more than one mode is present, however, individual r_min and r_max might be different, as one might want to "force" one mode into a specific subinterval.

*Line 218 misprint*

    A. Corrected

*Figure 2: axis labels?*

    A. Added

*All Figures: captions provide not enough information.*

    A. We added information in all captions.

*Only one CRI was used for the simulations: 1.49+0.019, which is known from other References that it is a "good" one, i.e., the degree of ill-posedness is small.*

    A. We thank the reviewer for their comment. We did test the method with several different values of CRI, but we are only using a single value in the manuscript to make the results easier to interpret. As the reviewer correctly indicates, the selected value is a "good one", although it is not an exceptionally good one (i.e., no cherry picking). As a rule of thumb, the retrieval becomes more difficult as the imaginary part of the CRI grows. The aim of the manuscript is to show that the method works fine when the data allow the retrieval; when the information content of the data

deteriorates, no method can provide good reconstructions unless additional sources of information are considered.

*Figure 8 is not discussed and the results are astonishing.*

    A. We thank the reviewer for this comment. We added a comment in the Results section

    "This result is particularly reasonable because all the datasets were indeed generated by bimodal distributions, including the one recorded on Etna which may be erroneously thought to correspond to a unimodal distribution."

    and we added reference to Figures 7-9 in the Discussion

*Section 3.4. Results with real data: This is not at all a retrieval with real data. This is only a simulation with size distributions and CRI which were found by AERONET retrievals.*

    A. We thank the reviewer for this comment. Indeed, as mentioned in a previous answer, we are not aware of the existence of lidar data where the true solution is known. In this case, we wanted to underline that these "real" data are the closest thing we could get to experimental measurements; however, we agree with the reviewer that this might be misleading: we now changed the wording to "quasi-real" to underline that they are not exactly experimental recordings, although they are based on them.

*Further References which are important and missing:*

*Ritter, et al., Microphysical Properties and Radiative Impact of an intense Biomass Burning aerosol event measured over Ny-Ålesund, Spitsbergen in July 2015, Tellus B: Chemical and Physical Meteorology, 2018.*

*Ortiz-Amezcua, et al., Microphysical characterization of long-range transported biomass burning particles from North America at three EARLINET stations, ACP, 2017.*

*Müller, et al., Microphysical particle properties derived from inversion algorithms developed in the frame of EARLINET, AMT, 2016.*

    A. References added, thanks for this.

---

## Author Response (AR3)

**Response Letter to the reviewers' comments on manuscript**

**"A Bayesian parametric approach to the retrieval of the atmospheric number size distribution from lidar data"**

In the following, the reviewers' comments are reported in italic, and the Authors' responses are marked with an "A." and given in plain text.

**Reviewer 1**

*The revised manuscript is improved and a lot of the reviewer questions are answered. But there are still a few open important points:*

   A. We thank the reviewer for this comment.

*- Only one CRI was used for the simulations: 1.49+0.019, which is known from other References that it is a "good" one, i.e., the degree of ill-posedness is small.*
*A: "… We did test the method with several different values of CRI, but we are only using a single value in the manuscript to make the results easier to interpret. … the retrieval becomes more difficult as the imaginary part of the CRI grows. …."*

*In my opinion this is a very critical point. The authors should at least show one more example with a more difficult, but of course realistic in nature, CRI and should give a remark to the limitation.*

   A. We thank the reviewer for this comment. In fact, the cases defined in the text "quasi-real" refer to SD determined by AERONET in three different sites, and were processed with the values of the refractive index determined by AERONET: therefore CRIs are different from the refractive index used for the simulations. But this was visible only in Table 8 and not explicitly reported and discussed in the text. We have therefore modified the text. Furthermore, in order to provide a term of comparison of the reconstruction of the same SDs with different CRI, we have added the results of the reconstruction of the same "quasi-real" cases with a rather extreme CRI value  of

1.57 + i0.043. This led to the addition of 3 figures (the (c) panels in Fig 7-9), a new table and the respective comment.

*- Abstract: We show that the proposed algorithm provides satisfactory results even when the assumed number of modes is different from the true number of modes, and substantially excellent results when the right number of modes is selected.*
*- Line 355: The preliminary results presented in this paper indicate that the proposed method can effectively retrieve uni– and bi–modal distributions from extinction coefficients measured at two wavelengths and backscattering coefficients measured at three wavelengths.*

*In contrast: Figure 6. Reconstructions obtained by running the inversion algorithm with a bimodal distribution, when data are generated by a bimodal distribution.*
*In contrast: Figure 8. Results obtained by applying our proposed method, with a unimodal (left) and with a bimodal (right) distribution, to the experimental dataset recorded on Etna.*

*In my opinion the two sentences (abstract and line 355) are far too optimistic!*

    A. We modified both sentences, that are now less optimistic.

*- Line 230: ….where r_min and r_max assume in our case the values 0.01μm and 20μm, respectively.*
*- Line 336: In the bimodal SD reconstruction, Fig. 6 and Table 6, the "fine" mode is always reconstructed better than the "coarse" mode, this effect is connected to the wavelengths used for determining the optical parameters of the lidar measures.*

*This is correct. But my main concern is the value of r_max=20μm. This could suggest that the algorithm is working even very well for such large particles. I do not believe this. In all examples or figures, respectively, one can see that r_max is only about 11μm which is already very large in comparison to the used wavelengths and only about the half of 20. Are there any examples made by the authors with a second mode between 10 and 20μm? This would be interesting for the community!*

    A. The value of r_max = 20mm refers only to the range used for the discretization of the radius values in the reconstruction; the maximum value of the modal radius of the SD that the method reconstructs is always less than 7μm in all the analyzed cases. We have modified the text to clarify this point.

**Reviewer 2**

*Authors answered correctly to many of my questions and the manuscript has been improved.*

    A. We thank the reviewer for this comment.

*However, I still have two concerns that need to be better discussed in the manuscript.*
*i) it is unrealistic to have Lidar measurements with less than 5 % errors in retrieved optical parameters. Better discussion is needed about the limitations in the retrieval with the proposed scheme.*

A. As reported by the referee, the error on the optical parameters plays a critical role in the quality of the reconstruction and the value of 5% on a single measurement is quite optimistic, especially in the measurement of the extinction coefficient. However, if we consider the average value of the optical parameters (2 alpha and 3 beta) corresponding to an atmospheric layer with a thickness of 1Km, with an error of 10-15% (see References [1,2] below) on a single point with a resolution of the order of 100m, the statistical error is less than 5%.

*ii) retrieval of aerosol refractive index is the key in current challenges in aerosol science, more even that size distribution. That is the weaker point of the proposed scheme. So further discussion is still needed in the manuscript*

A. We added a paragraph in the Discussion section where we discuss the limitation related to the assumption of known CRI:

"A second limitation concerns the subjectivity in the choice of the CRI, which was assumed to be known in the present study. Our analysis on quasi-real data showed how the quality of the retrieval may depend on the value of the CRI, and partly deteriorates when the imaginary part grows, particularly for larger modes. This is a known issue with lidar data, that can possibly be solved in a Bayesian framework by devising better priors. In addition, a full Bayesian model including the CRI among the unkonwns can be devised, however, with an increased number of unknowns it will be necessary to exploit more prior information to reduce the degree of ill-posedness."

[1] Mattis, Ina, et al. "Dual‑wavelength Raman lidar observations of the extinction‑to‑backscatter ratio of Saharan dust." Geophysical Research Letters 29.9 (2002): 20-1.

[2] GAW Aerosol Lidar Observation Network. "GAW Report No. 178."
https://library.wmo.int/doc_num.php?explnum_id=9387